# Formulation and In Vitro Efficacy Assessment of *Teucrium marum* Extract Loading Hyalurosomes Enriched with Tween 80 and Glycerol

**DOI:** 10.3390/nano12071096

**Published:** 2022-03-26

**Authors:** Mohammad Firoznezhad, Ines Castangia, Carlo Ignazio Giovanni Tuberoso, Filippo Cottiglia, Francesca Marongiu, Marco Porceddu, Iris Usach, Elvira Escribano-Ferrer, Maria Letizia Manca, Maria Manconi

**Affiliations:** 1Department of Pharmacy, University of Salerno, 84084 Fisciano, Italy; m_firoznejhad_20500@yahoo.com; 2Department of Scienze della Vita e dell’Ambiente, University of Cagliari, 09124 Cagliari, Italy; tuberoso@unica.it (C.I.G.T.); cottiglf@unica.it (F.C.); fmarongiu@unica.it (F.M.); porceddu.marco@unica.it (M.P.); mlmanca@unica.it (M.L.M.); manconi@unica.it (M.M.); 3Sardinian Germplasm Bank (BG-SAR), Hortus Botanicus Karalitanus (HBK), University of Cagliari, 09124 Cagliari, Italy; 4Department of Pharmacy and Pharmaceutical Technology and Parasitology, University of Valencia, Burjassot, 46100 Valencia, Spain; iris.usach@uv.es; 5Biopharmaceutics and Pharmacokinetics Unit, Institute for Nanoscience and Nanotechnology, University of Barcelona, 08028 Barcelona, Spain; eescribano@ub.edu

**Keywords:** plant extract, phospholipid vesicles, transfersomes, glycerosomes, antioxidant activity, fibroblasts, scratch assay, Mediterranean plant

## Abstract

The extract of *Teucrium marum* L. (Lamiaceae) was obtained using the aerial parts of the plant, by means of a maceration process. Verbascoside, caffeic acids derivatives and flavonols were the main components contained in the extract as detected using high-performance liquid chromatography coupled with diode array detector (HPLC–DAD) as an analytical method. The extract was successfully incorporated into hyalurosomes, which were further enriched by adding a water cosolvent (glycerol) and a surfactant (Tween 80), thus obtaining glycerohyalurosomes. Liposomes, transfersomes and glycerosomes were prepared as well and used as comparisons. All vesicles were small, as the mean diameter was never higher than ~115 nm, thus ideal for topical application and stable on storage, probably thanks to the highly negative surface charge of the vesicles (~−33 mV). The cryo-TEM images confirmed the formation of close-packed, oligolamellar and multicompartment hyalurosomes and glycerohyalurosomes in which around 95% of the used extract was retained, confirming their ability to simultaneously load a wide range of molecules having different chemical natures. Moreover, the extract, when loaded in hyalurosomes and glycerohyalurosomes was able to counteract the damages induced in the fibroblasts by hydrogen peroxide to a better extent (viability~110%) than that loaded in the other vesicles (viability~100%), and effectively promoted their proliferation and migration ensuring the healing of the wound performed in a cell monolayer (scratch assay) during 48 h of experiment. Overall in vitro results confirmed the potential of glycerohyalurosomes as delivery systems for *T. marum* extract for the treatment of skin lesions connected with oxidative stress.

## 1. Introduction

*Teucrium marum* (*T. marum*) is a plant belonging to the family of Lamiaceae, known as the largest family of flowering plants especially rich in essential oils [1,2]. According to the characteristics of the genus *Teucrium*, *T. marum* is an aromatic plant with a peculiar smell, so-called “mint plant” or sometimes “cat thyme”. It spontaneously grows in different habitats from sea level up to 2000 m altitude in the western Mediterranean islands and is endemic in Sardinia, Corsica, Tuscan Archipelago, Balearic and Hyères Islands [3,4,5,6]. This plant has been used since ancient times in folk medicine, for its antibacterial, anti-inflammatory and antipyretic activities [3]. In the northern part of Sardinia (Siniscola), it has been also used for the treatment of malaria [5]. These beneficial activities are mainly connected to the presence of diterpenoids, flavonoids and phenolic acids [7,8]. The plants belonging to the genus *Teucrium* are also known for their insect-repellent activity due to the content of more than 220 different diterpenes and especially neoclerodane diterpenes [9,10]. The essential oil of *T. marum*, containing sesquiterpenes such as caryophyllene oxide, α-bergamotene, β-bisabolene, β-caryophyllene, β-sesquiterphellandrene and iridoids, such as dolichodial, teucrein and dolicholactone [3], is widely employed to treat various diseases [3]. Some studies have been carried out to confirm the bacteriostatic, spasmolytic and anti-inflammatory effects of essential oil, which are probably related to the presence of several cyclopentanoid monoterpenes in the essential oil [5]. More recently, Ricci [5] and Eisner [11] underlined that some methylcyclopentanoid monoterpenes contained in *T. marum*, such as dolichodial and teucrein, exert an insecticidal activity and may be used as a defensive agent for plants.

The composition and efficacy of the crude extracts obtained by maceration of its aerial parts were poorly studied so far. Indeed, only one study reporting the chemical composition of some extracts from *T. marum* is reported in the literature [12]. However, its phytocomplex seem to be interesting and valuable from the biomedical point of view due to the presence of verbascoside [13]. Moreover, as reported for several phytocomplexes, the loading in ad hoc formulated nanocarriers may improve the local delivery and biological activities of payloads, decreasing the concentration needed to reach the beneficial and/or therapeutic effect [14]. Concerning the topical application, phospholipid vesicles seem to be the most suitable nanocarriers, due to their similarity to biological membranes, safety and ability to simultaneously load lipidic and hydrophilic molecules [15,16]. Several recent studies demonstrated the improvement of the therapeutic efficacy of different natural molecules by their nanoencapsulation in these lamellar vesicles [17,18,19,20]. They usually improve drug penetration and accumulation in intact skin, acting as both penetration enhancers and carriers [21,22,23].

Given that, in this study, the phytochemicals, from the aerial parts of *T. marum*, were extracted by maceration by using a mixture of water and ethanol as an extractive medium, and loaded in ad hoc formulated phospholipid vesicles. The phytocomplex was analyzed by means of the HPLC–DAD analytical method and loaded in liposomes, transfersomes, glycerosomes, hyalurosomes and hyalurosomes enriched with glycerol and Tween 80. The physicochemical (morphology, size, zeta potential, stability on storage) and technological (entrapment efficiency) properties of the vesicles were evaluated along with their biocompatibility, ability to protect fibroblasts from oxidative damage and capability of stimulating their proliferation and migration in vitro in a wounded cell monolayer.

## 2. Materials and Methods

### 2.1. Materials

Lipoid S75 (S75), a mixture of soybean phospholipids (~70% phosphatidylcholine, 9% phosphatidylethanolamine and 3% lysophosphatidylcholine), triglycerides and fatty acids, was purchased from AVG S.r.l. (Garbagnate Milanese, Milan, Italy), a local supplier for Lipoid GmbH (Ludwigshafen, Germany). Sodium hyaluronate was purchased from DSM Nutritional Products AG Branch Pentapharm (Aesch—BL/Switzerland). Tween 80, 2,2-diphenyl-1-picrylhydrazyl (DPPH), tetrazolium salt, 3-(4,5-dimethylthiazol-2-yl)-2,5-diphenyltetrazolium bromide (MTT), ethanol, acetonitrile, methanol, phosphoric acid, and all the other solvents and reagents of analytical grade were purchased from Sigma-Aldrich (Milan, Italy). Standards of luteolin-7-O-glucoside, luteolin, apigenin, chlorogenic acid and verbascoside were purchased from Extrasynthese (Genay Cedex, France).

Reagents and plastics for cell culture were purchased from Life Technologies Europe (Monza, Italia). Ultrapure water (18 MΩ·cm) was obtained using a Milli-Q Advantage A10 System apparatus (Millipore, Milan, Italy).

### 2.2. Plant Material Collection

The aerial parts of *T. marum* accessions were collected between May and July 2018 from Is Lisandrus (Buggerru), Su Logufresu (Uta), Monte Corrasi (Oliena), which are located in the south and the central north areas of Sardinia, respectively. A representative voucher specimen from each collection area was deposited in the Herbarium CAG of the University of Cagliari, Italy. The aerial parts of the plants were cleaned and dried for ten days at room temperature (23 ± 5 °C), then were powdered by using a grinder, to obtain a powder composed of small particles and a high superficial area.

### 2.3. Extraction Process

The dried and powdered aerial parts of *T. marum* (100 g) were macerated using 500 mL of a mixture of ethanol and water (70:30) as extractive medium. The dispersion was left under constant stirring (200 rpm) for 72 h, at room temperature (25 °C), and sonicated for 5 min every 24 h to improve the extraction yield. After 72 h, the extractive dispersion was centrifuged twice (30 min, 8000 rpm) to remove the coarse material. The obtained extract was then diluted with water (1:100) and lyophilized by using a freeze-drier, Operon FDU8606 (Nuova Criotecnica Amcota, Rome, Italy), to obtain a green/dark powder, which was vacuum packed in a dark glass container until its use [24,25].

### 2.4. HPLC–DAD Analysis of the Extract

The quantitative analysis of the extract was carried out using a modified HPLC–DAD method as previously described by Gil et al. [26]. An Agilent Technologies 1260 Infinity HPLC system fitted with a pump module G7111, an autosampler module G7129A (10 µL injection volume) and a G4212B photodiode array detector (Agilent Technologies, Cernusco sul Naviglio, Milan, Italy) was used for the analyses. The separation was obtained with a Gemini C18 column (150 × 4.60 mm, 3 µm, Phenomenex, Casalecchio di Reno, Bologna, Italy) using 0.22 M phosphoric acid (solvent A) and acetonitrile (solvent B) as mobile phase at a constant flow rate of 1.0 mL/min. A gradient was generated by keeping 100% of solvent A up to 5 min, then decreasing the solvent A up to 85% in 15 min, up to 65% in 20 min, up to 10% in 50 min, and again up to 100% in 55 min. The chromatograms and spectra were processed with OpenLab V. 2.51 software (Agilent Technologies, Cernusco sul Naviglio, Milan, Italy). The detection of the different molecules was performed at different wavelengths: 360 nm for flavonols, 313 nm for hydroxycinnamic acids, 520 nm for anthocyanins, and 280 nm for all the other phenolics. Individual components were identified by comparing the retention time and UV–vis spectra of pure commercial standards or the UV–vis spectra and the chromatographic profile described in the literature. Stock standard solutions of chlorogenic acid, verbascoside, luteolin-7-O-glucoside, luteolin and apigenin were prepared in methanol and the working standards in ultrapure water. The calibration curves were obtained using the external standard method, correlating the area of the peaks with the concentration. Correlation values were 0.9997 in the range of 2–50 mg/L for the three standards. The *T. marum* extract was dissolved in methanol (40 mg/mL), diluted 1:50 *v*/*v* with a mixture of methanol:water 80:20 *v*/*v*, filtered through a GD/X cellulose acetate membrane (0.20 μm, Ø 25 mm, Whatman, Milan, Italy) and injected into the HPLC without any further purification. The data were expressed in mg/g of dry extract.

### 2.5. Vesicle Preparation

Soy lecithin (S75, 180 mg/mL) and the dried extract (40 mg/mL) were weighed in a glass vial and hydrated with water to obtain liposomes. Tween 80 (15 mg/mL) was added to the mixture composed of phospholipid and extract and hydrated with water to obtain transfersomes. The same solid mixture was hydrated with a blend of glycerol and water (25:75) to obtain glycerosomes and with sodium hyaluronate in water to prepare hyalurosomes, which were further modified by adding glycerol to the sodium hyaluronate aqueous solution (25:75), thus obtaining glycerohyalurosomes. The dispersions were immediately sonicated (25 cycles, 5 s on and 2 s off) by using a Soniprep 150 ultrasonic disintegrator (MSE Crowley, London, UK) to obtain small and homogeneous systems. The composition of the samples is reported in Table 1.

### 2.6. Vesicle Characterization

Vesicle formation and morphology were evaluated by cryo-TEM analysis. A thin film of each sample was formed on a holey carbon grid and vitrified by plunging (kept at 100% humidity and room temperature) into ethane maintained at its melting point, using a Vitrobot (FEI Company, Eindhoven, The Netherlands). The vitreous films were transferred to a Tecnai F20 TEM (FEI Company), and the samples were observed in a low-dose mode. Images were acquired at 200kV at a temperature of ~−173 °C, using a CCD Eagle camera (FEI Company).

The average diameter and polydispersity index (a measure of the width of the size distribution) of the vesicles were determined by dynamic and electrophoretic light scattering using a Zetasizer ultra (Malvern Instruments, Worcestershire, UK). The zeta potential was estimated using the same Zetasizer ultra, which converts the electrophoretic mobility by means of the Smoluchowski approximation of the Henry equation. Before the analysis, samples were diluted (1:100) with water to be optically clear and avoid the attenuation of the laser beam by the particles along with the reduction of the scattered light that can be detected, then analyzed at 25 °C [27].

To evaluate the amount of the phytocomplex incorporated into the vesicles, samples (2 mL) were purified from the non-incorporated components by dialyzing (Spectra/Por^®^ membranes: 12–14kDa MW cut-off, 3 nm pore size; Spectrum Laboratories Inc., DG Breda, The Netherlands) them against water (4 L) for 2 h, refreshing the water after 1 h. The amount of water used (8 litres in total), the maintenance of the system under constant stirring and the change of the medium aiming at ensuring the sink conditions, were suitable for the complete removal of non-incorporated bioactives. At the end of the purification process, the antioxidant activity (AA) of the extract into the vesicles, before and after dialysis, was evaluated by measuring their ability to scavenge 2,2-diphenyl-1-picrylhydrazyl (DPPH) radical. Each sample was diluted (1:400) with a DPPH methanolic solution (40 μg/mL) and was stored at room temperature, in the dark, for 30 min to allow the reaction between the different components. Then the absorbance was measured at 517 nm against blank. All the experiments were performed in triplicate. The entrapment efficiency (EE), which is the ability of the samples to retain the bioactives inside them, was calculated as a percentage of the antioxidant activity found after dialysis versus that detected in the unpurified samples (before dialysis).

A stability study was carried out as well, by monitoring the average size, the polydispersity index and the surface charge of the vesicles, stored at room temperature (~25 ± 1 °C) for a period of 120 days.

### 2.7. DPPH• Assay

The antioxidant activity of the extracts solubilized in methanol or loaded in vesicles was assessed by measuring their ability to scavenge the DPPH• radical. Each sample was previously diluted (1:10) to reduce the colour of the resulting solution and to ensure a clear and reliable absorbance reading, then the samples (50 μL) were dissolved in 1950 μL of DPPH• methanolic solution (40 μg/mL), to keep the hydrophobic hydrazyl radical in solution while offering sufficient buffering capacity, which in turn ensures the complete reaction between the different components during 30 min of incubation in the dark, at room temperature. Then, the absorbance was measured at 517 nm against blank. All the experiments were performed in triplicate. The antioxidant activity was calculated according to the formula: antioxidant activity % = [(ABSDPPH − ABSsample)/ABSDPPH] × 100 [28].

### 2.8. Biocompatibility of the Formulations

Mouse embryonic fibroblasts (3T3) (ATCC collection, Manassas, VA, US) were incubated at 37 °C, 100% humidity and 5% CO_2_ and grown as monolayers in 75 cm^2^ flasks. Dulbecco’s Modified Eagle Medium with high glucose, supplemented with fetal bovine serum (10%), penicillin, streptomycin and fungizone, was used to culture the cells. Cell viability was measured using 3-[4,5-dimethylthiazol-2-yl]-3,5 diphenyl tetrazolium bromide (MTT) colourimetric test. Briefly, cells (7.5 × 10^3^ cells/well) were seeded in 96-well plates and after 24 h were treated with 25 μL of formulations opportunely diluted with medium to reach the desired concentrations of the extracts (0.2, 2, 20, 40 μg/mL). After 48 h of incubation, cells were washed with PBS, treated with MTT solution (100 μL, 0.5 mg/mL, final concentration), which was replaced after 3 h with dimethyl sulfoxide (100 μL). Finally, the absorbance of the solubilized dye was read at 570 nm by using a microplate reader (MultiskanEX, Thermo Fisher Scientific, Inc., Waltham, MA, US). Results are expressed as the percentage of viability of treated cells versus that of untreated control cells (100% cell viability) [29,30].

### 2.9. Protective Effect of the Formulations against Cell Damages Induced by Hydrogen Peroxide

Mouse embryonic fibroblasts (3T3) were seeded into 96-well plates and incubated at 37 °C in 5% CO_2_ for 24 h. Then, cells were stressed with hydrogen peroxide (25 μL, 30% solution diluted 1:40,000), and immediately incubated for 4 h with 25 μL of the extract in dispersion or loaded in vesicles (2 μg/mL of extract, final concentration). Untreated cells (100% of viability), and cells stressed with hydrogen peroxide only, were used as positive and negative control, respectively. After 4 h, cells were washed with phosphate buffer solution and the cell viability was measured using the MTT test as described above (Section 2.8), [31].

### 2.10. Scratch Assay

The ability of the formulations to stimulate both migration and proliferation of fibroblast was evaluated by means of the scratch assay test [32,33]. Cells were seeded in 6-well plates and kept in culture until the complete confluence was reached. Subsequently, a thin wound was generated on the cell monolayer using a sterile plastic tip. Cell fragments were gently removed by washing each well with the medium preheated at 37 °C. Immediately after the generation of the wound (time 0), cells were treated with the extract in dispersion or loaded in vesicles (2 mg/mL of extract, final concentration) and incubated for up to 48 h. Untreated cells were used as a negative control. At each time point (0, 8, 24, 32 and 48 h) cells were observed using an optical microscope (10× objective) to detect both cell proliferation/migration and the closing speed of the wounded area [32,33].

### 2.11. Statistical Analysis of Data

Results are expressed as means ± standard deviations. Analysis of variance (ANOVA) was used to evaluate multiple comparisons of means and Tukey’s test and Student’s *t*-test were performed to substantiate differences between groups using XL Statistics for Windows. The differences were considered statistically significant for *p* < 0.05.

## 3. Results

### 3.1. Extraction of the Bioactives and Characterization of the Extract

The aerial parts of *T. marum* were dried at room temperature and lyophilized in order to remove the remaining water and facilitate the grinding process and consequently the extraction yield. At the end of the ultrasound-assisted maceration, the extractive dispersion was centrifuged, diluted with water (1:100) and freeze-dried. The lyophilised powder was then stored under vacuum until its use.

From a qualitative point of view, the HPLC–DAD analysis revealed a very high level of caffeic acid derivatives and flavonols in the extract. Anthocyanins have not been identified and only traces of benzoic acid were detected in the chromatogram at 280 nm (data not shown). Figure 1 reports the chromatograms recorded at 313 and 360 nm. From the study of the UV–vis spectra (200–600 nm) and by comparison with pure standards or literature data, a total of 12 compounds related to hydroxycinnamic acids and 14 related to flavonols were detected. Hydroxycinnamic acids were mainly represented by chlorogenic acid (an ester of caffeic acid and quinic acid) and verbascoside (a caffeoyl phenylethanoid glycoside). An unknown compound with a UV–vis spectrum very similar to that of verbascoside (matching > 99.8%) was detected at 23.245 min.

The main components of extract were quantified (Table 2) and its total phenolic content was 221.72 ± 8.44 mg/g dr. The extract was characterized by a high content of verbascoside (48.50 ± 0.49 mg/g dr), followed by chlorogenic acid (20.26 ± 0.10 mg/g dr) and luteolin-7-O-glucoside (16.67 ± 0.19 mg/g dr). No quantitative data on the phenolic composition of *T. marum* can be found in the literature so far, but the detected compounds have been reported in other Teucrium species [7,34].

### 3.2. Vesicle Preparation and Characterization

The prepared vesicles were observed using the cryo-TEM to confirm the actual formation of closed and lamellar vesicles and to know their structure and morphology (Figure 2).

Transfersomes were small, homogeneously dispersed (PI ~ 0.22) and mostly unilamellar, some small and elongated vesicles were observed as well (Figure 2A). The addition of glycerol improved both sphericity and lamellarity as the vesicles were more spherical and similar to each other in terms of size, as confirmed by the lower value of the polydispersity index (~0.16) (Figure 2B), the lamellarity of these vesicles also increased leading the formation of oligolamellar systems, as previously reported [22]. The structure of hyalurosomes was even different, as they appeared spherical, oligolamellar and close-packed, probably due to the presence of sodium hyaluronate that may immobilize and maintain the vesicles in dispersion (Figure 2C) [30,32]. A small amount of large vesicles were also detected (PI ~ 0.25), with a peculiar multicompartment organization characterized by larger vesicles involving other smaller vesicles inside them, which may be responsible for the improved wound healing effect, indeed these peculiar systems may control, to a better extent than the other vesicles, the release of the payload in the site of application. The addition of glycerol and Tween 80, further modify the morphology of the vesicles, as they were less aggregated, and mainly composed of large and multicompartment structures slightly polydispersed [35,36,37].

To confirm overall cryo-TEM results, size, polydispersity index and zeta potential of the vesicles were measured by means of dynamic and electrophoretic light scattering (Table 3). All prepared vesicles were small in size and mostly monodispersed, with a polydispersity index < 0.25.

Any significant difference was detected among the mean diameters of liposomes, transfersomes, glycerosomes and hyalurosomes (*p* > 0.05, among the values). Only the mean diameter of glycerohyalurosomes was larger than the others, as confirmed by cryo-TEM analyses. All vesicles were highly negatively charged due to the negative charge of phosphatidylcholine. Moreover, vesicles incorporated ~95% of the used extract without significant differences between samples, confirming their ability to simultaneously load various molecules having different chemical natures.

Average size, polydispersity index and zeta potential of the vesicles were monitored for 120 days of storage at ~25 °C (Figure 3). The vesicular dispersions were stable as any change higher than 10% was detected up to 120 days for the measured values.

The antioxidant activity of *T. marum* extract loaded in vesicles was measured and compared to that of the extract dissolved in methanol at the same concentration. The antioxidant activity of the free extract in solution was ~90% and increased up to ~130% when it was loaded in vesicles, probably due to the antioxidant power of soybean phospholipids associated with the other components of the extract (Table 3).

### 3.3. Biocompatibility of the Formulations

The biocompatibility of the *T. marum* extract in dispersion or loaded in vesicles was evaluated using fibroblasts, as these cells can be considered the most representative of the deeper skin layer, i.e., the dermis. The cells were incubated for 48 h with the extract loaded in vesicles, and with the extract dispersed in an aqueous solution, which was used as a comparison to evaluate the effect of carriers. Samples were diluted with the cell medium to reach different concentrations of extract (40, 20, 2 and 0.2 mg/mL) and then, the viability of cells was measured as a function of their metabolic activity using the MTT test (Figure 4).

The viability of the extract in dispersion or loaded in liposomes was ~88% (*p* > 0.05 between the viability of cells treated with the two samples) and was slightly higher for transfersomes (viability ~93%), irrespective of the used concentration, indicating that the extract was highly biocompatible. The viability of cells treated with the extract loaded transfersomes at higher dilutions slightly increased up to ~100% (*p* < 0.05 versus the viability of cells treated with the extract dispersion), that of cells treated with glycerosomes significantly increased (~116%) (*p* < 0.05 versus the viability of cells treated with the extract loaded in liposomes and transfersomes), and a further increase was detected (~125%) when hyalurosomes and glycerohyalurosomes at the lower dilution were used. The improved biocompatibility of extract loaded hyalurosomes can be related to the presence of sodium hyaluronate, which is capable of stimulating fibroblast proliferation [38]. Any significant difference was detected as a function of the concentration used.

### 3.4. Protective Effect of Formulations against Damages Induced by Hydrogen Peroxide in Fibroblasts

The ability of formulations to protect fibroblasts against oxidative stress induced by hydrogen peroxide was evaluated as well (Figure 5). Hydrogen peroxide is a dangerous molecule capable of inducing apoptosis and death of cells [39]. As expected, its addition in cell medium significantly reduced fibroblast viability up to ~42% (*p* < 0.05 versus the other values) [40]. The viability of cells treated with the extract in dispersion increased up to ~90%, disclosing the high quality of the obtained extract, which was biocompatible and capable of counteracting oxidative stress of cells. The protective effect of the extract was slightly increased by its loading in vesicles up to ~100% but without statistically significant differences between samples (*p* > 0.05 among the values provided by dispersion and liposomes, transfersome, glycerosomes and glycerohyalurosomes). Only the incubation with the extract loaded hyalurosomes achieved a higher cell viability (~110%, *p* < 0.05 versus other values) with respect to that measured for cells treated with extract in dispersion or loaded in the other vesicles. Overall results confirmed the safety and effectiveness of these formulations, especially hyalurosomes, and their possible application in the treatment of skin disorders associated with oxidative stress.

### 3.5. Effect of Formulations on Proliferation and Migration of Fibroblasts

In vitro tests performed using cells have been widely used in the last decades as they are reliable and can be considered predictive of the in vivo effects, but particularly they are preferred, especially to avoid the use of animal models. In light of this, the ability of *T. marum* extract in dispersion or loaded in vesicles, to promote cell proliferation and migration has been evaluated using the scratch assay. The test consists of the simulation of a skin wound in a cell monolayer and on the evaluation of the healing ability of the formulations [33,36], which was measured over 48 h at scheduled time points (Figure 6).

A monolayer of control cells was wounded and untreated with formulations to observe the physiological remodelling behaviour of fibroblasts. In this case, the closure of the lesion was slow and detectable only at 36 and 48 h. The treatment of the wounded monolayer with the extract in dispersion slightly stimulated the proliferation and migration of cells even if at 48 h its closure was still incomplete. Similar behaviour was observed using liposomes and transfersomes as the closure of the wounded area was not reached at 48 h but seems to be more evident already at 24 h in comparison with that of cells treated with the extract in dispersion. The treatment with glycerosomes led to better results and at 48 h the wound closure was almost complete. The complete closure at 48 h was obtained by treating the cells with hyalurosomes and glycerohyalurosomes; especially when using hyalurosomes, an almost closed wound was observed already at 24 h. According to the results reported above, the extract loaded glycerosomes, hyalurosomes and glycerohyalurosomes significantly promoted the proliferation of fibroblasts but only extract loaded hyalurosomes seemed to be really effective in promoting the cell migration in the wounded area.

## 4. Discussion

The obtained extract seems to have an interesting and promising composition due to its high content of verbascoside, a caffeoyl phenylethanoid glycoside compound, also known as acteoside [39]. This molecule is known to have antioxidant, anti-inflammatory and photoprotective effects especially when applied to the skin [41,42]. Georgive et al. found that the fraction of the crude extract of *Verbascum xanthophoeniceum* Griseb. containing verbascoside effectively restored the altered chemokine expression in normal human keratinocytes [43]. Its wound healing efficacy was previously confirmed by de Maroua Sperotto et al., which demonstrated that *Plantago australis* Lam. hydroethanolic extract, standardized in verbascoside, promoted wound healing in both in vitro and in vivo models [44]. Moreover, Ambrosone et al., for the first time, loaded verbascoside in liposomes and used them as an effective delivery system for corneal wound treatment [45]. Considering these previous promising results, the *T. marum* extract rich in verbascoside was loaded in different phospholipid vesicles to select the most suitable and capable of improving its wound healing effectiveness. Indeed, it was previously demonstrated that the beneficial properties of each phytocomplex or phytochemical can be maximized by selecting the most suitable delivery systems [16]. Phospholipid vesicles were chosen because of their versatility and their ability to simultaneously load both hydrophilic and lipophilic compounds, thus being ideal carriers for the incorporation of plant-derived phytocomplexes, containing different bioactive molecules [46]. Hyalurosomes were selected due to their optimal performance and effectiveness on skin lesions, and they were further enriched by using Tween 80, a surfactant, which can act as an edge activator, and glycerol, which has an emollient and moisturizing effect on the skin [47]. Liposomes, transfersomes and glycerosomes were used as comparisons. The vesicles were prepared by direct sonication of the dispersions, thus avoiding the use of organic solvents [48,49,50]. The morphology of prepared vesicles was significantly affected by their composition as transfersomes were mainly unilamellar and small, the addition of glycerol improved the lamellarity of vesicles and the addition of sodium hyaluronate immobilized the vesicles, which appeared close-packed and multicompartment [36]. In particular, the presence of sodium hyaluronate or even the combination of the polymer with a surfactant (Tween 80) and a cosolvent (glycerol), promoted the formation of larger vesicles with smaller ones inside them, which can be responsible for the better interaction with the cells and the control of the release of the bioactive inside them. Even if the structure and morphology were dependent on the composition of the vesicles, their physicochemical properties were similar, as reported above. Indeed, all samples were small in size and homogeneously dispersed, only hyalurosomes and glycerohyalurosomes were both slightly larger and polydispersed, probably due to a different bilayer assembling, also confirmed by cryo-TEM, linked to the presence of additives with different properties [51]. As these systems have been specifically tailored to be used for the treatment of skin disorders, their biocompatibility was evaluated using fibroblasts, which provide a valuable model to evaluate the effectiveness of a specific drug and to reduce animal testing [52]. The extract, in dispersion or loaded in vesicles, was not toxic, as the viability of fibroblasts was never lower than 90%, in addition, hyalurosomes and glycerohyalurosomes at lower dilutions improved the cell viability up to ~110%, and in the scratch assay effectively stimulated both cell migrations and wound closure, probably due to a synergic effect of the sodium hyaluronate contained in these formulations [38]. Indeed, hyaluronic acid is a physiological biopolymer, which significantly contributed to the increased proliferation and spreading of fibroblasts, thus favouring wound healing [53]. In addition, it has also antioxidant activity and in this study, corroborated the antioxidant activity of *T. marum* extract maximizing the ability of the extract loaded hyalurosomes to counteract the damages induced in fibroblast by hydrogen peroxide [54]. This formulation was the most effective, also in comparison with glycerohyalurosomes, which provided lower protection against hydrogen peroxide damages, which can be related to the different structure and morphology of vesicles. We can hypothesize that the close-packed structure of hyalurosomes facilitates their interaction with the fibroblasts and the internalization of the phytochemicals. Inside the cells, they can counteract to a better extent the lipidic and DNA damages caused by hydrogen peroxide and even address proliferation [30]. An additional advantage of this formulation is the increased viscosity, with respect to liposomes or transfersomes, which avoid leakage of bioactives at the application site, thus improving both interaction with the skin and penetration in the deeper strata [55].

## 5. Conclusions

Overall results disclosed that the maceration in water–ethanol of aerial parts of *T. marum* is a suitable method to obtain an extract rich in verbascoside and other antioxidant polyphenols. The phytochemical can be incorporated in liposomes, transfersomes, glycerosomes, hyalurosomes and glycerohyalurosomes using an easy and scalable preparation method. Hyalurosomes seem to be the most effective vesicles for the treatment of skin lesions even if all the formulations are biocompatible, capable of protecting fibroblasts from oxidative stress and promoting their proliferation and migration, suggesting their use for the prevention and/or treatment of skin diseases connected with oxidative stress.

## Figures and Tables

**Figure 1 nanomaterials-12-01096-f001:**
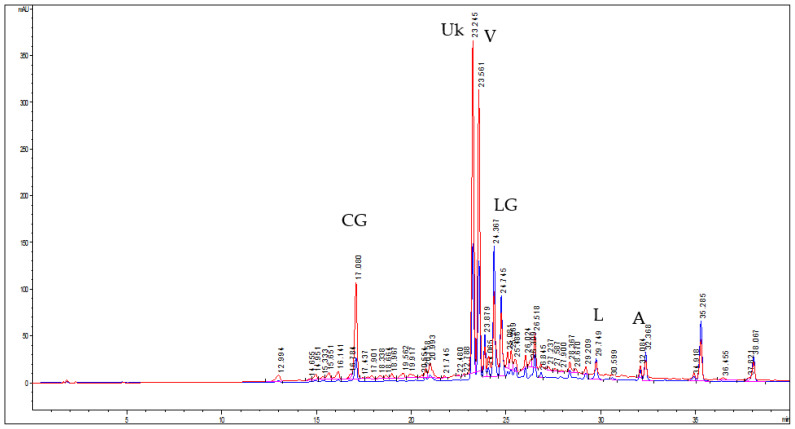
Chromatograms at 313 nm (red line) and 360 nm (blue line) of *T. marum* extract obtained by ultrasound-assisted maceration in ethanol and water (70:30), where the peaks of chlorogenic acid (CGA), unknown (Uk), verbascoside (V), luteolin-7-O-glucoside(LG), luteolin (L), apigenin (A) are presented.

**Figure 2 nanomaterials-12-01096-f002:**
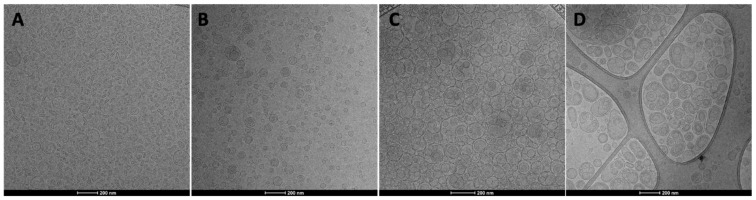
Representative cryo-TEM images of transfersomes (**A**), glycerosomes (**B**), hyalurosomes (**C**) and glycerohyalurosomes (**D**).

**Figure 3 nanomaterials-12-01096-f003:**
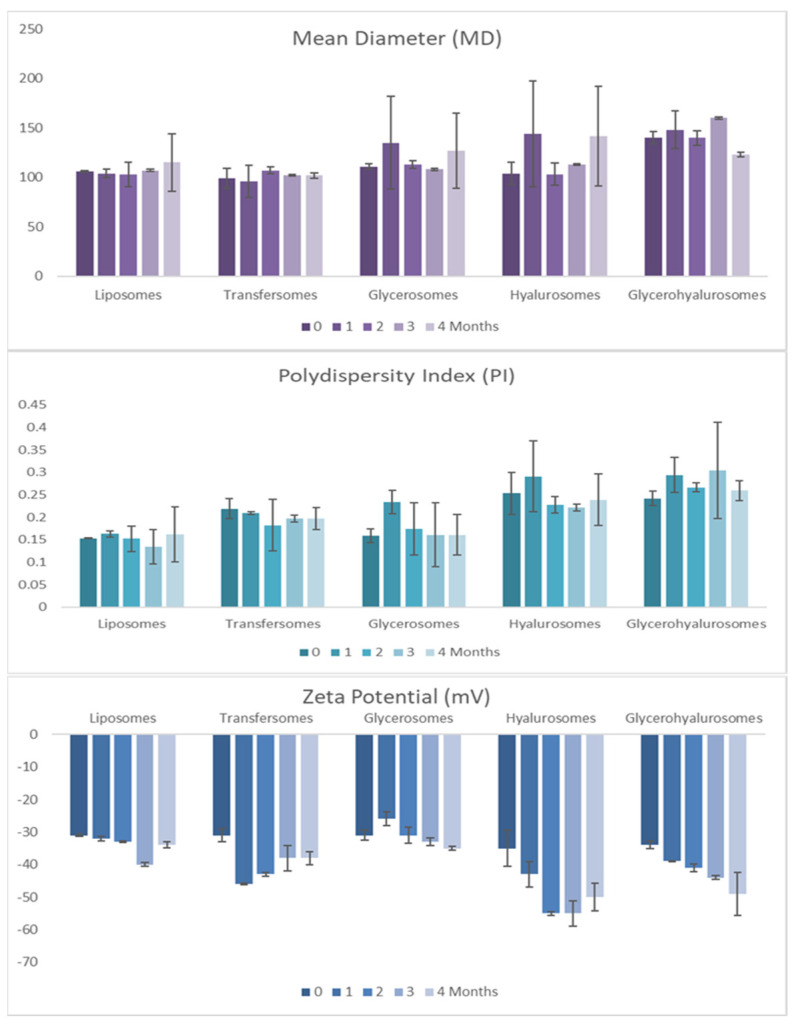
Mean diameter (MD), polydispersity index (PI) and zeta potential (ZP) of vesicles loading *T. marum* extract over 120 days of storage.

**Figure 4 nanomaterials-12-01096-f004:**
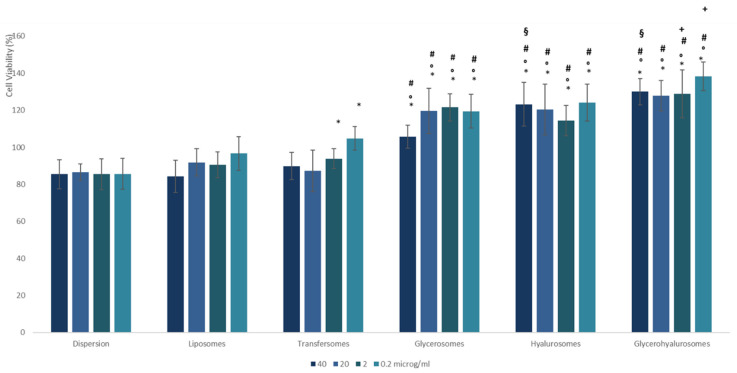
Viability of fibroblasts incubated for 48 h with the *T. marum* extract in dispersion or loaded in vesicles. Mean values ± standard deviations are reported. The symbol * indicates values statistically different from the extract in dispersion (*p* < 0.05); the symbol ° indicates values statistically different from liposomes (*p* < 0.01); the symbol # indicates values statistically different from transfersomes (*p* < 0.01); the symbol § indicates values statistically different from glycerosomes (*p* < 0.05) and the symbol + indicates values statistically different from hyalurosomes (*p* < 0.01).

**Figure 5 nanomaterials-12-01096-f005:**
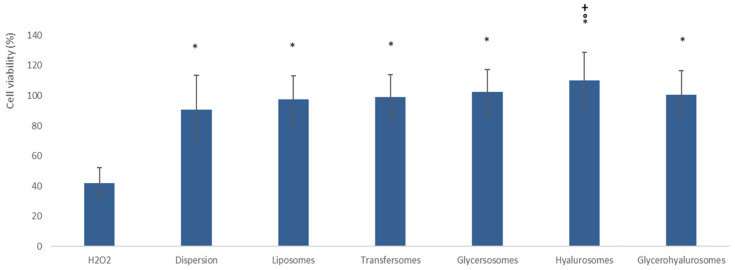
Viability of fibroblasts stressed with hydrogen peroxide and treated with *T. marum* extract in dispersion or loaded in vesicles. Mean values ± standard deviations are reported. The symbol * indicates values that were statistically different from hydrogen peroxide (*p* < 0.01); the symbol ° indicates values that were statistically different from the dispersion of the extract (*p* < 0.05); and the symbol + indicates values that were statistically different from liposomes (*p* < 0.05).

**Figure 6 nanomaterials-12-01096-f006:**
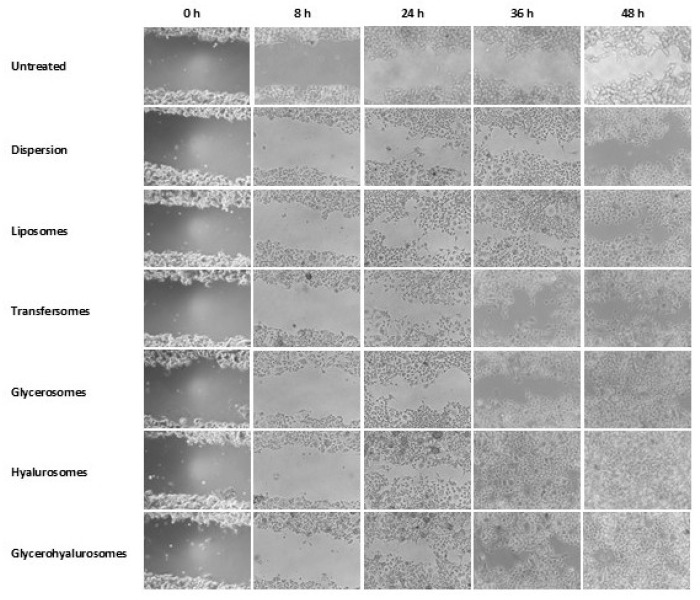
Representative images of the scratch performed in fibroblast monolayers and treated with the extract in dispersion or loaded in vesicles.

**Table 1 nanomaterials-12-01096-t001:** Composition of *Teucrium marum* dried extract loaded vesicles.

	S75	Extract	Tween 80	Sodium Hyaluronate	Glycerol	Water
	mg	mg	mg	mg	mL	mL
Liposomes	180	40	-	-	-	1
Transfersomes	180	40	15	-	-	1
Glycerosomes	180	40	-	-	0.25	0.75
Hyalurosomes	180	40	-	5	-	1
Glycerohyalurosomes	180	40	15	5	0.25	0.75

**Table 2 nanomaterials-12-01096-t002:** The main components detected and quantified in the extract of *T. marum* obtained by ultrasound-assisted maceration in ethanol and water (70:30). The unknown phenol was quantified using the calibration curve of verbascoside, others after verbascoside using the curve of chlorogenic acid and others after apigenin using the curve of luteolin-7-O-glucoside. Mean values ± standard deviations are reported (*n* =3).

*T. marum* Extract	mg/g dr
Total phenolic compounds	221.72 ± 8.44
Hydroxycinnamic acids	151.70 ± 2.46
Chlorogenic acid	20.26 ± 0.10
Unknown	58.97 ± 0.52
Verbascoside	48.50 ± 0.49
Others	34.45 ± 1.57
Flavonols	70.02 ± 0.83
Luteolin-7-O-glucoside	16.67 ± 0.19
Luteolin	3.56 ± 0.04
Apigenin	2.01 ± 0.01
Others	47.78 ± 0.44

**Table 3 nanomaterials-12-01096-t003:** Mean diameter (MD), polydispersity index (PI), zeta potential (ZP), entrapment efficiency (EE%) and antioxidant activity (AA%) of *T. marum* extract loaded in liposomes, transfersomes, glycerosomes, hyalurosomes and glycerohyalurosomes. Mean values ± standard deviations are reported (*n* ≥ 6).

	MD (nm)	PI	ZP (mV)	EE%	AA%
Liposomes	106 ± 11	0.15	−31 ± 3	95 ± 2	95 ± 3
Transfersomes	99 ± 9	0.22	−31 ± 5	96 ± 5	136 ± 9
Glycerosomes	111 ± 13	0.16	−31 ± 7	97 ± 4	97 ± 4
Hyalurosomes	104 ± 7	0.25	−35 ± 6	99 ± 3	100 ± 6
Glycerohyalurosomes	140 ± 15	0.24	−34 ± 4	98 ± 2	98 ± 5

## Data Availability

Not applicable.

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
