# Peer review of "Formulation and In Vitro Efficacy Assessment of Teucrium marum Extract Loading Hyalurosomes Enriched with Tween 80 and Glycerol"

_nanomaterials, 2022, doi:10.3390/nano12071096_

Round 1
Reviewer 1 Report
Reviewer 1
The manuscript “Formulation and in vitro efficacy assessment of hyaluro-2 somes containing with tween 80 and glycerol and loading 3 Teucrium marum extract” submitted in Nanomaterials Journal is quite interesting with scientific approach to treat the skin lesions.In my opinion, this manuscript is premature at this moment and required extensive revision.
- The manuscript is required extensive English or grammar correction and there is room for improvement.
- Some results/data has to be incorporated in abstract for scientific purpose as well reader point.
- When I am looking table 3, how the EE% is 99±3, 97±4, 98±2 and s on reported of the different formulation. Author does not mention how they performed the EE, particle size, PDI, as these characterization is very important and crucial for the vesicle formulation. A exhaustive methodology for characterization is required.
Author Response
We would like to thank the Reviewers for the remarks and suggestions that have allowed us to improve the quality of the paper.
A point-to-point list of answers follows. The amended text is shown in red in the revised manuscript.

Reviewer 2 Report
The authors formulated various T. Marum extract-loaded nanocarriers for the treatment of skin lesions. The in vitro results confirmed the potential of glycerohyalurosomes in the effective delivery of the T. marum extract for the treatment of skin lesions compared to the other nanostructures. The nanostructure characterization and visualization should be improved. The morphology has a particular impact on the healing of skin lesions, therefore not only the average particle size but the size distribution is important. The authors should provide the histograms of the particle distributions of various nanostructures. Thus the structure-activity relationship should be analyzed from the point of the morphology of the particles. I propose the publication after minor revision (addition of particle size distributions along with their discussion regarding the activity).
Author Response

(The authors gave the same response as above.)

Round 2
Reviewer 1 Report
The revised manuscript Formulation and in vitro efficacy assessment of Teucrium marum extract loading hyalurosomes enriched with tween 80 and glycerol is quite improved and revised well by the authors and may be acceptable
In my opinion the quality of the fig 3,4 and5 need to be improved.
Author Response
We would like to thank the Reviewers for the remarks and suggestions that have allowed us to improve the quality of the paper.
